# Biopsychosocial stressors and perinatal mental health: The mediating role of social support in a Pakistani cohort

Abid Malik[1] , Rakhshanda Liaqat[1,2] , Nadia Suleman[1], Semra Etyemez[3], Bilal Ahmed[1], Raima Asif[1], Shahzad Ali Khan[1], Pamela J. Surkan[4] and Lauren M. Osborne[5]

[1]Health Services Academy, Pakistan; [2]Human Development Research Foundation, Pakistan; [3]Weill Cornell Medicine, USA; [4]Johns Hopkins Bloomberg School of Public Health, USA and [5]Departments of Obstetrics and Gynaecology and Department of Psychiatry, Weill Cornell Medical College, USA

## Research Article

**Keywords:**
Anxiety; biomarkers; depression; developing countries; perinatal

**Corresponding author:**
Abid Malik;
Email: abid.malik@hsa.edu.pk

## Abstract

Perinatal depression and anxiety are major contributors to maternal morbidity, with a disproportionate burden in low- and middle-income countries. In Pakistan, common and modifiable biological risks, including anemia and vitamin D deficiency, may interact with psychosocial factors to influence perinatal mental health. This cohort study enrolled 152 pregnant women from a public hospital in Islamabad; 147 completed baseline assessments (12–32 weeks gestation) and 100 were followed at 6–8 weeks postpartum. Validated Urdu versions of the EPDS, GAD-7, and MSPSS were used alongside hemoglobin and vitamin D assessments at both time points. Longitudinal analyses were conducted using generalized linear mixed models, supplemented by cross-sectional and mediation analyses. Depression was prevalent antenatally (41.5%) and increased postpartum (57.0%), while anxiety declined from 25.2% to 12.0%. Higher hemoglobin was protective against antenatal depression (OR = 0.66) and anxiety (OR = 0.65), but not in longitudinal models. Vitamin D deficiency predicted postnatal depression (OR = 3.15), while sufficiency was associated with remission. Social support showed a strong protective effect (OR = 0.24) and mediated 40% of the hemoglobin–depression association. Baseline symptom severity was the strongest predictor of postpartum outcomes. These findings highlight a substantial burden and point to modifiable nutritional and psychosocial targets for intervention.

## Impact statement

Perinatal common mental health disorders remain major yet under-recognized public health concerns in low- and middle-income countries (LMICs), where women face a dual burden of biological vulnerability and social adversity. This longitudinal cohort study provides rare integrated evidence from Pakistan demonstrating that readily measurable and highly prevalent nutritional deficiencies, particularly low hemoglobin and vitamin D levels, significantly influence perinatal mental health, and that social support both directly protects mothers and mediates biological risk. These findings highlight actionable and scalable pathways for early identification and intervention through existing maternal health services. By demonstrating that simple biomarkers routinely collected in antenatal care can be used to stratify mental health risk, and that strengthening social support can mitigate biological vulnerabilities, this work has direct relevance for global maternal health policy, clinical practice and implementation science. The study offers a pragmatic biopsychosocial model that can inform integrated, low-cost maternal mental health strategies in LMICs and other resource-constrained settings worldwide.

## Introduction

Perinatal common mental disorders are highly prevalent and disabling conditions, now recognized as significant global public health challenges (Howard and Khalifeh, 2020). Perinatal depression and anxiety often co-occur but are distinct disorders with different etiologies and clinical outcomes (Kalin, 2020). Globally, major depression affects about 12% of women during pregnancy or after childbirth. In comparison, anxiety symptoms affect an estimated 10–15% (Fawcett et al., 2019). Both conditions disrupt maternal well-being, interfere with maternal infant bonding and increase the risk of negative developmental outcomes for children, including emotional, behavioral and cognitive difficulties (Imran et al., 2021). Meta-analyses from Pakistan suggest that nearly 37% of women experience depression during pregnancy, with

postnatal rates remaining around 30% (Atif et al., 2021; Waqas et al., 2023), suggesting this to be a public health issue of priority. Anxiety is also common, with prevalence estimates ranging between 18% and 30%, and often co-occurring with depression (Runkle et al., 2023). These high levels of distress are shaped by structural and social factors such as poverty, gender inequality, family pressures and limited access to mental health care (Nazir et al., 2022a).

The perinatal period is a biologically sensitive time, marked by significant immunological, metabolic and neuroendocrine changes (Saraswat et al., 2021). These shifts can increase vulnerability to mental health problems but also offer opportunities to study underlying mechanisms (Osborne et al., 2019). Research in high-income countries has examined biological contributors across several systems, including epigenetic regulation, neuroactive steroid pathways, immune activation and hypothalamic–pituitary–adrenal (HPA) axis dysregulation (Osborne *et al.* 2025). For instance, dysregulation of placental corticotropin-releasing hormone predicts postpartum depressive symptoms (Glynn and Sandman, 2014). More recent work has identified changes in neuroactive steroid pathways, such as enzymes involved in 3α-HSD metabolism, as potential predictors of postpartum depression (Osborne et al., 2025). Roles for HPA axis dysfunction and immune pathways have also been suggested in perinatal anxiety (Ceruso et al., 2020). However, this evidence base is drawn almost entirely from Western populations. Very few studies have evaluated biological mechanisms of perinatal depression and anxiety in LMICs, and work from South Asia remains particularly limited. In Pakistan, our group recently reported elevated chemokine activity across pregnancy among clinically anxious women (Etyemez et al., 2025). The biological underpinnings of perinatal depression in this context are largely unexamined.

Nutritional deficiencies are one such overlooked factor (Asim et al., 2022). In Pakistan, anemia affects nearly half of all pregnant women (Habib et al., 2020), and vitamin D deficiency is also widespread (Al-Jawaldeh et al., 2024). Both conditions have biological links to mental health (Abiri and Vafa, 2020). Anemia can cause fatigue, low energy and cognitive problems, which may increase vulnerability to depression and anxiety (Fekih-Romdhane and Jahrami, 2023). Vitamin D plays an essential role in neurotransmission, immune regulation and brain development (Mahar et al., 2024a). Low vitamin D levels have been consistently linked to depression, including in perinatal populations, although evidence for their association with anxiety is less clear (Zhang et al., 2020). Very few studies, however, have tracked how these deficiencies influence the course of depression and anxiety over time, especially in LMICs (Herba et al., 2016). Importantly, both hemoglobin and vitamin D are routinely assessed within antenatal care in Pakistan, making them pragmatic candidates for investigating biologically relevant correlates of perinatal mental health in low-resource settings.

Psychosocial resources also play a central role in shaping mental health outcomes. Perceived social support is one of the strongest protective factors against perinatal depression (O'Hara and McCabe, 2013), and studies from Pakistan confirm that women with higher levels of social support report fewer depressive symptoms (Waqas et al., 2024). While social support's effect on perinatal anxiety is less studied, evidence suggests a protective role here as well (Harrison et al., 2020). The "weathering" hypothesis helps explain this: chronic exposure to stress and social adversity gradually undermines health, increasing susceptibility to both physical and psychological problems (Simons et al., 2021). Nutritional

deficiencies may interact with these psychosocial pathways (Mirza et al., 2018). For example, a woman with anemia may struggle with fatigue, making it harder to participate in her social networks, which in turn reduces the support she receives (Van Der Woude et al., 2014). However, the specific ways in which biological and social risk factors co-occur or interact in relation to depression *versus* anxiety across the perinatal period remain poorly understood, particularly in longitudinal designs.

Significant methodological and contextual gaps persist in the extant literature. A primary limitation is the predominant reliance on cross-sectional data, which obscures how biological and psychosocial factors relate to changes in perinatal mental health symptoms over time, leaving it unresolved whether biomarkers precede, co-occur with or are consequences of the psychiatric condition (Osborne and Monk, 2013). Furthermore, the concurrent integration of biological and psychosocial variables within multivariate models remains scarce; this siloed approach impedes a nuanced understanding of their potential synergistic or mediating relationships in the etiology of perinatal mental illness (Duberstein et al., 2021). This knowledge gap is most acute in LMICs. For instance, there is a pronounced scarcity of longitudinal research incorporating biological risk factor assessments in settings like Pakistan, which concurrently experiences a high prevalence of both maternal nutritional deficiencies and perinatal depression (Insan, 2023). While foundational work from Pakistan has effectively identified key psychosocial determinants of mental health conditions, such as intimate partner violence and low social support (Malik et al., 2024), the systematic incorporation of objective biological measures, such as inflammatory cytokines or nutrient levels, into these psychosocial frameworks has been exceptionally limited (Kendall-Tackett, 2007).

To address these gaps, this longitudinal cohort study in Pakistan was designed to investigate the intersection of biological vulnerabilities and psychosocial resources in shaping perinatal mental health. We focused on two prevalent and modifiable nutritional deficiencies, anemia and vitamin D deficiency. We examined their association with the trajectories of depression and anxiety from pregnancy to the postpartum period. We also integrated perceived social support, assessing its role both as an independent protective factor and as a potential mediator between biological risk factors and psychological outcomes. By integrating biological and psychosocial dimensions, this study aims to advance a more holistic understanding of perinatal mental health in a high-need, understudied population and to inform future research and contextually appropriate strategies that consider both nutritional status and psychosocial resources among mothers in Pakistan.

## Methods

### Study design and setting

This study employed a prospective longitudinal cohort design to examine psychosocial and biological predictors of perinatal depression and anxiety. It was reported in accordance with the Strengthening the Reporting of Observational Studies in Epidemiology guidelines for cohort studies (von Elm et al., 2007). Data collection was conducted between January 2024 and January 2025 at two time points: baseline during pregnancy (T1: 12–32 weeks' gestation) and follow-up postpartum (T2: 6–8 weeks after delivery). Participants were recruited from the Gynecology and Obstetrics outpatient department (OPD) of the Federal General Hospital (FGH),

Islamabad. The department serves ~180 women daily and provides both outpatient and inpatient obstetric care under the supervision of consultant obstetricians, postgraduate residents and trained nursing staff. Women attending routine antenatal checkups were approached during their waiting time in the OPD, and psychosocial assessments were conducted after obtaining informed consent. Biological samples were collected at the Islamabad Diagnostic Centre (IDC), a nationally accredited laboratory, following standardized biospecimen handling procedures.

### Participants and recruitment

The sample size was determined for this prospective cohort study to examine the association between low vitamin D levels and postpartum depression (PPD). In the absence of local data on perinatal mental disorders and nutritional biomarkers in Pakistan, the calculation was based on a prior Iranian cohort that reported a 32% higher PPD prevalence among women with low vitamin D (Abedi et al., 2018). Assuming a PPD prevalence of 15% and 47% in women with normal and low vitamin D, respectively, a total of 152 participants was required. This calculation used a standard formula for comparing two proportions, with a two-sided alpha of 0.05, 80% power and a 15% inflation to account for anticipated attrition.

Participants were consecutively recruited during routine antenatal visits at the Gynecology and Obstetrics Outpatient Department. Eligible women were 18–45 years old, currently pregnant (12–32 weeks of gestation) and had lived within the hospital's catchment area for at least 6 months. Exclusion criteria included

severe chronic medical conditions (*e.g.*, uncontrolled diabetes, hypertension or cardiac disease requiring hospitalization), current obstetric complications (*e.g.*, preeclampsia) or a known active psychiatric disorder, such as psychotic or bipolar disorder.

Written informed consent was obtained in the Urdu language. For illiterate participants, study details were read aloud, and a close relative witnessed consent. A total of 152 women were enrolled at baseline, of whom 147 completed both psychosocial and biological assessments. At 6–8 weeks postpartum, 100 participants were retained for follow-up, resulting in a 68% retention rate (Figure 1).

### Measures

#### Demographic and clinical variables

Comprehensive demographic and clinical data were collected at baseline, including maternal age, parity, educational attainment, occupation, household structure and medical history. Anthropometric measurements (height and weight) were obtained using standardized protocols to calculate body mass index (WHO expert, 2004). BMI was classified according to the World Health Organization guidelines.

Depressive symptoms. Depressive symptoms were measured at both time points using the Edinburgh Postnatal Depression Scale (Cox et al., 1987). The EPDS is a 10-item self-report questionnaire specifically designed to screen for perinatal depression. Items are rated on a 4-point scale (0–3) based on symptoms experienced over the past 7 days, with total scores ranging from 0 to 30. Higher scores indicate greater depressive symptom severity. In this study, we used

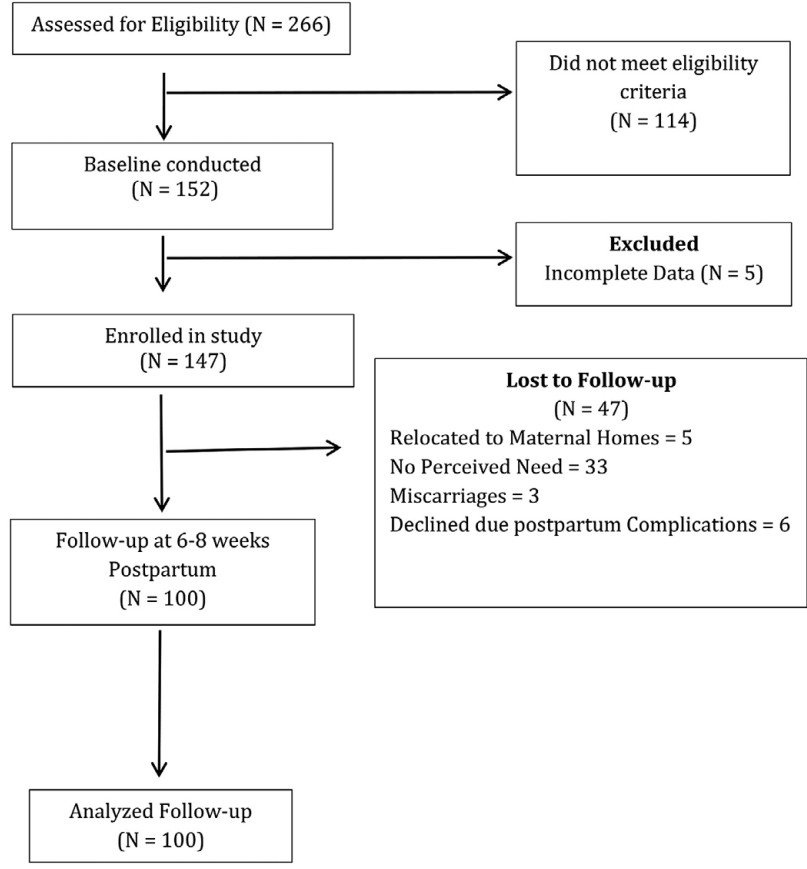

**Figure 1.** Participant flow.

both continuous scores and a categorical classification with a cutoff score of ≥13 indicating depression (Cox et al., 1987). The Urdu version of the EPDS, validated in Pakistani populations (Dosani et al., 2022), was used to assess maternal depressive symptoms during the perinatal period.

**Anxiety symptoms.** Anxiety symptoms were assessed at both time points using the Generalized Anxiety Disorder 7 (GAD-7)-item scale (Spitzer et al., 2006). This self-report measure asks respondents to rate how frequently they experienced core anxiety symptoms over the past 2 weeks using a 4-point Likert scale ranging from 0 (not at all) to 3 (nearly every day). Total scores range from 0 to 21, with higher scores indicating greater anxiety severity. We analyzed scores both continuously and categorically, using the established cutoff of ≥10 to identify cases of moderate to severe anxiety (Spitzer et al., 2006). The Urdu version of the GAD-7, previously validated in Pakistan (Ahmad et al., 2017), was used to assess maternal anxiety symptoms during the perinatal period.

**Perceived social support.** Social support was measured at both time points using the Multidimensional Scale of Perceived Social Support (Zimet et al., 1988). The scale consists of 12 items that assess perceived support from family, friends and significant others. In this study, items were rated on a 5-point Likert scale (Maselko et al., 2015) with responses ranging from 1 (very strongly disagree) to 5 (very strongly agree). Higher total scores indicate a greater perceived social support. Total scores were analyzed as a continuous variable.

**Biomarkers.** Following psychosocial assessments at the FGH, participants were accompanied by trained study coordinators to the adjacent IDC branch for the collection of biological samples. At the IDC facility, venous blood samples (20 mL) were collected at both time points by certified phlebotomists using standardized aseptic procedures. Samples were immediately transported under controlled cold-chain conditions (maintained below $-20$ °C) and processed within 2 hours of collection to ensure sample integrity. Analyses were performed according to established laboratory protocols to determine hemoglobin and 25-hydroxyvitamin D [25(OH) D] concentrations. Standard international cutoff values were applied: hemoglobin levels below 12 g/dL were classified as anemia; vitamin D status was categorized as *deficient* (<20 ng/mL), *insufficient* (20–29 ng/mL) or *sufficient* (≥30 ng/mL). All laboratory analyses were conducted in accordance with certified quality control procedures (PNAC Islamabad, 2024). Biomarker results were double-entered into a secure electronic database and linked with psychosocial assessment data using unique participant identification codes by trained research assistants.

### Statistical analysis

All analyses were performed in R (version 4.3.1). Descriptive statistics, including means, standard deviations, frequencies and percentages, were used to characterize the sample at baseline (T1) and follow-up (T2). Cross-sectional associations at each time point were examined using logistic regression models with depression (EPDS ≥13) and anxiety (GAD-7 ≥ 10) as binary outcomes.

To assess potential bias due to loss to follow-up, baseline characteristics of participants retained at the postnatal assessment (T2) were compared with those lost to follow-up. Comparisons included sociodemographic variables (maternal age, education, parity and family structure), biological markers (hemoglobin and vitamin D), perceived social support (MSPSS) and baseline mental health symptom severity (EPDS and GAD-7 scores). Independent samples *t*-tests were used for continuous variables, and $\chi^2$ or Fisher's exact tests were applied for categorical variables, as appropriate. These analyses were conducted to evaluate whether attrition was systematically related to exposure, mediator or outcome variables.

To examine the longitudinal course of symptoms, trajectory groups were derived based on participants' diagnostic status at both antenatal (T1) and postnatal (T2) assessments using standard EPDS and GAD-7 cutoffs. Trajectories were classified as persistent (clinical symptoms at both T1 and T2), postnatal-onset (nonclinical at T1, clinical at T2), remitted (clinical at T1, nonclinical at T2) and healthy (nonclinical at both time points). Associations between trajectory groups and baseline biomarker status (hemoglobin and vitamin D) were examined using Fisher's exact tests. Given small subgroup sizes, these analyses were interpreted as exploratory. Where overall associations were statistically significant, post-hoc pairwise comparisons were conducted.

An exploratory mediation analysis was conducted using baseline (T1) data to examine whether perceived social support (MSPSS) was statistically associated with the relationship between biological markers and mental health symptoms. Analyses were performed using the mediation package in R with 5,000 bootstrap samples to estimate indirect, direct and total effects, as well as the proportion of the association statistically accounted for by the mediator. As exposure, mediator and outcome variables were measured concurrently, these analyses were interpreted as associative rather than causal.

Longitudinal predictors across the perinatal period were assessed using mixed-effects models. Generalized linear mixed models (GLMMs) with a binomial distribution and logit link were used for binary outcomes (EPDS ≥ 13 and GAD-7 ≥ 10), and linear mixed models were used for continuous symptom scores (EPDS, GAD-7 and MSPSS). All models included a random intercept for participants to account for within-subject correlation. Baseline symptom severity was included in longitudinal models due to its established predictive importance for postnatal outcomes. Due to the reduced sample size at follow-up, longitudinal and trajectory analyses had limited statistical power. A complete-case analysis approach was used for all analyses, and model diagnostics and sensitivity checks were conducted.

This study was supported by the International Brain Research Organization through the Neuroscience Capacity Accelerator for Mental Health (Application ID: IW-6827570252).

### Ethical considerations

Ethical approval was obtained from the National Bioethics Committee of Pakistan, Islamabad, and the Institutional Review Board (IRB) of Health Services Academy. The study prioritized participant welfare: informed consent procedures were meticulously followed, confidentiality was ensured through data anonymization, and referral pathways to psychiatric and obstetric services were established for participants with severe symptoms, nutritional deficiencies or disclosures of violence.

### Results

### Characteristics of study participants

At baseline (T1), 147 women completed psychosocial and biomarker assessments. Participants were relatively young (mean

age = 26.1 years, SD = 4.6), with most not employed (90.6%) and the majority living in joint or extended family households (80.5%). Educational levels varied: 12.1% had never attended school, while nearly half (45.6%) had completed matriculation 10th grade. High burdens of deficiency were observed, with 87.8% of women anemic and 81.0% vitamin D deficient. Clinically significant symptoms were also common, with 41.5% meeting the threshold for probable depression and 25.2% for anxiety. Full sociodemographic, clinical and biomarker details are presented in Table 1.

### Attrition analysis

A total of 147 women completed baseline assessments; of these, 100 (68%) were retained at follow-up and 47 (32%) were lost to follow-up. Table 2 presents a comparison of baseline characteristics between participants retained at T2 and those lost to follow-up.

No statistically significant differences were observed between groups with respect to maternal age, employment status, family structure, educational attainment, anemia prevalence, vitamin D deficiency or baseline depression and anxiety status (all $p > 0.05$). These findings suggest that attrition was not systematically associated with baseline sociodemographic factors, nutritional biomarkers or mental health symptom severity. Despite minimal attrition bias, the reduced follow-up sample limits the statistical power for detecting associations in trajectory and longitudinal analyses.

### Biomarker associations with mental health trajectories

Analysis of mental health trajectories revealed distinct patterns by disorder. A significant association was found between vitamin D status and depression trajectory groups ($p = 0.02$, Fisher's exact test). As shown in Table 3, women whose antenatal depression remitted after childbirth had a markedly higher proportion of vitamin D sufficiency (50.0%) compared to those with persistent (7.7%) or postnatal-onset (16.1%) depression. Pairwise comparisons confirmed that the remitted group differed significantly from all other trajectory groups (all $p < 0.05$). In contrast, no significant association was observed between vitamin D status and anxiety trajectories ($p = 0.09$); therefore, pairwise comparisons were not conducted. Hemoglobin status was not associated with trajectory membership for either disorder (all $p > 0.05$). Given the small subgroup sizes within several trajectory categories, particularly for anxiety, this trajectory analyses are exploratory in nature and intended to be descriptive rather than inferential. The lack of association between hemoglobin and symptom trajectories is consistent with findings from the adjusted longitudinal mixed-effects models, in which hemoglobin did not independently predict postnatal mental health outcomes.

### Cross-sectional associations at baseline (12–32 weeks of gestation)

Cross-sectional analyses at baseline (T1) identified significant protective factors (Table 4). Each unit increase in hemoglobin was associated with ~35% lower odds of both depression (OR = 0.66, 95% CI: 0.49–0.86.) and anxiety (OR = 0.65, 95% CI: 0.47–0.88). Social support demonstrated an even stronger association, with higher scores linked to 76% lower odds of depression (OR = 0.24, 95% CI: 0.11–0.45) and 69% lower odds of anxiety (OR = 0.31, 95%

**Table 1.** Baseline characteristics of study participants ($N = 149$)

| Characteristic | Category | n (%)/Mean ± SD |
|---|---|---|
| **Sociodemographic** | | |
| Family structure | Nuclear | 28 (18.8) |
| | Joint | 66 (44.3) |
| | Extended | 54 (36.2) |
| | Multiple households | 1 (0.7) |
| Participant employment | Employed | 14 (9.4) |
| | Not employed | 135 (90.6) |
| Husband's employment | Employed | 141 (94.6) |
| | Not employed | 8 (5.4) |
| Participant education | No schooling | 18 (12.1) |
| | Primary/Middle school | 40 (26.9) |
| | Matric (10 years) | 68 (45.6) |
| | Intermediate (12 years) | 21 (14.1) |
| Husband's education | No schooling | 17 (11.4) |
| | Primary/Middle school | 25 (16.8) |
| | Matric (10 years) | 69 (46.3) |
| | Intermediate (12 years) | 17 (11.4) |
| | Graduate and above | 21 (14.1) |
| **Maternal traits** | | |
| BMI category (kg/m$^2$) | Underweight | 5 (3.4) |
| | Normal | 90 (60.4) |
| | Overweight | 37 (24.8) |
| | Obese | 15 (10.1) |
| Gestational age (weeks) | | 24.9 ± 6.7 (12–32) |
| Delivery mode ($N = 106$) | Normal delivery | 63 (59.4) |
| | Cesarean section | 44 (41.5) |
| Birth outcome ($N = 106$) | Preterm birth | 14 (13.1) |
| **Mental health and psychosocial** | | |
| EPDS score | | 11.10 ± 5.69 |
| Depression (EPDS ≥13) | | 63 (41.5) |
| GAD–7 score | | 6.31 ± 4.61 |
| Anxiety (GAD–7 ≥ 10) | | 39 (25.2) |
| MSPSS score (social support) | | 3.91 ± 0.76 |
| Biomarkers | | |
| Hemoglobin (g/dL) | | 10.56 ± 1.27 |
| Hemoglobin deficiency (<12 g/dL) | | 129 (87.8) |
| Vitamin D (ng/mL) | | 14.42 ± 6.76 |
| Vitamin D deficiency (<20 ng/mL) | | 119 (81.0) |

*Note*: Sociodemographic data were collected at baseline ($N = 149$). $N$s for mental health, psychosocial and biomarker variables vary due to missing values (*e.g.*, EPDS, GAD-7, Hb $N = 147$). Delivery mode and birth outcomes were obtained at follow-up ($N = 106$) and are therefore reported separately from baseline characteristics.

**Table 2.** Attrition analysis: comparison of baseline characteristics between participants retained and lost to follow-up

| Characteristic | Retained at T2 (*n* = 100) | Lost to follow-up (*n* = 47) | *p*-value |
|---|---|---|---|
| Age (years), mean ± SD | 26.3 ± 4.5 | 25.7 ± 4.8 | 0.48 |
| Employed, *n* (%) | 10 (10.0) | 4 (8.5) | 0.79 |
| Joint/extended family, *n* (%) | 82 (82.0) | 36 (76.6) | 0.46 |
| Education, *n* (%) | | | 0.88 |
| No formal schooling | 11 (11.0) | 7 (14.9) | |
| Up to matriculation | 47 (47.0) | 20 (42.6) | |
| Intermediate or above | 42 (42.0) | 17 (36.2) | |
| Anemia (Hb <12 g/dL), *n* (%) | 87 (87.0) | 42 (89.4) | 0.69 |
| Vitamin D deficiency (<20 ng/mL), *n* (%) | 82 (82.0) | 37 (78.7) | 0.64 |
| Depression (EPDS ≥13), *n* (%) | 43 (43.0) | 18 (38.3) | 0.58 |
| Anxiety (GAD–7 ≥ 10), *n* (%) | 26 (26.0) | 11 (23.4) | 0.73 |

*Note*: Values are presented as mean ± SD or *n* (%). *p*-values were derived using independent samples *t*-tests for continuous variables and $\chi^2$ or Fisher's exact tests for categorical variables, as appropriate.

CI: 0.16–0.53). Vitamin D deficiency was associated with significantly higher odds of anxiety (OR = 0.34, 95% CI: 0.14–0.83).

### Cross-sectional associations at follow-up (6–8 weeks postpartum)

At follow-up (T2), 57.0% of women met the threshold for probable depression (EPDS ≥13), while 12.0% met the threshold for anxiety (GAD-7 ≥ 10). The protective associations observed at baseline were not maintained in the postnatal period (Table 5). At T2, neither hemoglobin nor social support predicted postpartum

**Table 4.** Cross-sectional associations between biomarkers, social support and mental health at baseline (T1; *N* = 147)

| Outcome | Exposure | Estimate (95% CI) | *p*-value |
|---|---|---|---|
| Depression | Hemoglobin | OR = 0.66 (0.49–0.86) | 0.003 |
| Anxiety | Hemoglobin | OR = 0.65 (0.47–0.88) | 0.006 |
| Depression | Social support | OR = 0.24 (0.11–0.45) | <0.001 |
| Anxiety | Social support | OR = 0.31 (0.16–0.53) | <0.001 |
| Depression | Vitamin D deficiency | OR = 0.45 (0.19–1.02) | 0.059 |
| Anxiety | Vitamin D deficiency | OR = 0.34 (0.14–0.83) | 0.016 |

*Note*: Results are from logistic regression analyses presented as odds ratios (OR) with 95% confidence intervals.

depression or anxiety. However, vitamin D deficiency emerged as a significant risk factor for postnatal depression, with deficient women having over three times higher odds of depression (OR = 3.15, 95% CI: 1.13–9.44).

### Longitudinal biomarker associations with mental health

To identify independent predictors of perinatal mental health trajectories, we constructed multivariable GLMM for depression and anxiety, adjusting for key baseline covariates. The results are presented in Table 6. For depression, the odds of having clinically significant symptoms (EPDS ≥ 13) were nearly three times higher in the postpartum period than during pregnancy (OR = 2.82, 95% confidence interval [CI]: 1.48–5.38). The strongest predictor of postnatal depression was the baseline EPDS score (OR = 1.34, 95% CI: 1.24–1.45). In these adjusted longitudinal models, neither hemoglobin nor vitamin D levels at baseline were significant independent predictors of depression trajectory. This indicates that baseline depressive symptom severity was the dominant predictor of postnatal depression in the adjusted models.

**Table 3.** Vitamin D status and statistical associations with perinatal mental health trajectories (*N* = 100)

| (A) Depression trajectories | | | | |
|---|---|---|---|---|
| Trajectory group | Total *n* | Vitamin D normal *n* (%) | Vitamin D deficient *n* (%) | Statistical comparison (*p*) |
| Postnatal-onset depression | 31 | 5 (16.1) | 26 (83.9) | *vs.* Remitted: 0.029* |
| No depression (healthy) | 29 | 5 (17.2) | 24 (82.8) | *vs.* Remitted: 0.035* |
| Persistent depression | 26 | 2 (7.7) | 24 (92.3) | *vs.* Remitted: 0.004** |
| Antenatal-only (remitted) | 14 | 7 (50.0) | 7 (50.0) | Reference group |
| Overall association | | | | p = 0.02* |
| (B) Anxiety trajectories | | | | |
| Trajectory group | Total *n* | Vitamin D normal *n* (%) | Vitamin D deficient *n* (%) | |
| Postnatal-onset anxiety | 9 | 1 (11.1) | 8 (88.9) | |
| No anxiety (healthy) | 61 | 8 (13.1) | 53 (86.9) | |
| Persistent anxiety | 3 | 1 (33.3) | 2 (66.7) | |
| Antenatal-only (remitted) | 27 | 9 (33.3) | 18 (66.7) | |
| Overall association | | | | p = 0.09 (NS) |

*Note*: Trajectory groups were classified as follows: *Persistent* (clinical symptoms at both T1 and T2), *Postnatal-onset* (nonclinical symptoms at T1, clinical symptoms at T2), *Remitted* (clinical symptoms at T1, nonclinical symptoms at T2) and *Healthy* (nonclinical symptoms at both T1 and T2). *p*-values were derived from Fisher's exact tests. Pairwise comparisons were conducted only for depression trajectories where the overall test was significant.

**Table 5.** Cross-sectional associations between biomarkers, social support and mental health at follow-up (T2; *N* = 100)

| Outcome | Exposure | Estimate (95% CI) | *p*-value |
|---|---|---|---|
| Depression | Hemoglobin | OR = 0.97 (0.71–1.34) | 0.874 |
| Anxiety | Hemoglobin | OR = 0.97 (0.61–1.59) | 0.912 |
| Depression | MSPSS | OR = 1.04 (0.63–1.70) | 0.873 |
| Anxiety | MSPSS | OR = 1.35 (0.63–3.80) | 0.505 |
| Depression | Vitamin D deficiency | OR = 3.15 (1.13–9.44) | 0.032 |
| Anxiety | Vitamin D deficiency | OR = 1.23 (0.29–8.51) | 0.804 |

*Note*: Results are from logistic regression analyses presented as odds ratios (OR) with 95% confidence intervals.

**Table 6.** Factors associated with perinatal depression and anxiety in longitudinal analysis

| Outcome | Exposure | Estimate (95% CI) | *p* |
|---|---|---|---|
| Depression (EPDS ≥ 13) | Time (T2) | OR = 2.82 (1.48–5.38) | 0.002 |
| | Hemoglobin | OR = 1.02 (0.77–1.35) | 0.891 |
| | Vitamin D | OR = 0.97 (0.93–1.02) | 0.233 |
| | Gestational age | OR = 0.99 (0.94–1.04) | 0.653 |
| | Baseline EPDS | OR = 1.34 (1.24–1.45) | <0.001 |
| Anxiety (GAD–7 ≥ 10) | Time (T2) | OR = 0.21 (0.08–0.53) | 0.001 |
| | Hemoglobin | OR = 1.04 (0.74–1.46) | 0.831 |
| | Vitamin D | OR = 1.02 (0.96–1.08) | 0.514 |
| | Gestational age | OR = 1.02 (0.95–1.09) | 0.581 |
| | Baseline GAD–7 | OR = 1.50 (1.33–1.69) | <0.001 |

*Note*: Results from generalized linear mixed models (GLMMs). All models were adjusted for time point, baseline hemoglobin, baseline vitamin D, gestational age and baseline symptom score (continuous EPDS/GAD-7).

For anxiety, a different pattern emerged. The odds of clinically significant anxiety (GAD-7 ≥ 10) were significantly lower post-partum than during pregnancy (OR = 0.21, 95% CI: 0.08–0.53). Similar to depression, the baseline anxiety score was the strongest predictor of postnatal anxiety (OR = 1.50, 95% CI: 1.33–1.69). Hemoglobin and vitamin D levels were not significant predictors of anxiety in the adjusted models.

## Mediation analysis

An exploratory mediation analysis was conducted to examine whether perceived social support (MSPSS) was statistically associated with the relationship between hemoglobin levels and

**Table 7.** Mediation analysis of hemoglobin → social support → depression

| Effect | Estimate (95% CI) | *p*-value |
|---|---|---|
| ACME (indirect) | −0.37 (−0.72 to −0.09) | 0.006 |
| ADE (direct) | −0.54 (−1.22 to 0.13) | 0.112 |
| Total effect | −0.92 (−1.64 to −0.21) | 0.012 |
| Proportion mediated | 0.40 (0.09 to 1.33) | 0.016 |

*Note*: ACME, average causal mediation effect; ADE, average direct effect. Estimates are unstandardized.

depressive symptoms at baseline (T1). The results are presented in Table 7 and Figure 2. The indirect effect associated with perceived social support was statistically significant (ACME = −0.37, 95% CI: −0.72 to −0.09), indicating that lower hemoglobin levels were associated with higher depressive symptom scores, with this association statistically linked to lower perceived social support. The direct effect of hemoglobin on depressive symptoms was not statistically significant (ADE = −0.54, 95% CI: −1.22 to 0.13), while the total effect remained statistically significant (−0.92, 95% CI: −1.64 to −0.21). Approximately 40% of the total association between hemoglobin levels and depressive symptoms was statistically accounted for by perceived social support.

Mediation analyses examining other pathways (hemoglobin-anxiety, vitamin D-depression and vitamin D-anxiety) were not statistically significant and are therefore not discussed further.

## Discussion

This longitudinal cohort study provides novel, integrated biopsychosocial evidence on perinatal mental health from Pakistan. We documented an exceptionally high burden of symptomatology, with depressive symptoms affecting 41.5% of women antenatally and 57.0% postnatally, and anxiety symptoms affecting 25.2% and 12.0%, respectively. These elevated prevalences are consistent with those seen in LMICs but exceed global averages (Vidyasagaran et al., 2023). Furthermore, this high prevalence may be exacerbated by recent contextual stressors in Pakistan, including the post-COVID-19 pandemic recovery period, significant economic hardships and the devastating impacts of widespread flooding, which have collectively increased psychosocial and financial strain on families. Additionally, our urban hospital-based sample may include a significant number of internal migrants who have moved to Islamabad for work, potentially resulting in smaller, less established local social support networks, thereby increasing their vulnerability to perinatal distress.

Our findings reveal several key patterns that highlight the complexity of perinatal mental health. First, we observed distinct temporal trajectories for depression and anxiety, with the odds of

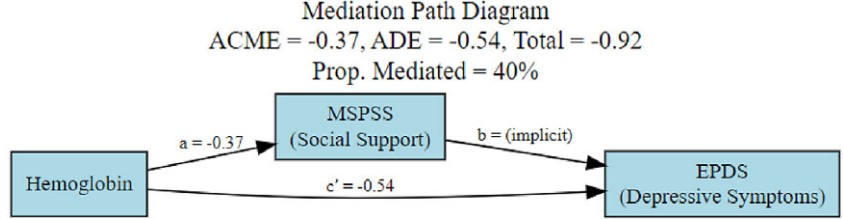

**Figure 2.** Mediation path diagram.

depression significantly increasing from pregnancy to postpartum, while the odds of anxiety decreased. Second, the relationship between nutritional biomarkers and mental health was highly context-dependent. Hemoglobin demonstrated a consistent protective association against antenatal depression and anxiety in cross-sectional analysis, and this association was statistically linked to perceived social support. However, in longitudinal models adjusting for baseline symptoms, hemoglobin was not a significant independent predictor of symptom trajectories. Third, vitamin D exhibited time-sensitive effects, with deficiency acting as a risk factor for postnatal depression and sufficiency being strongly associated with remission from antenatal depression. Fourth, and perhaps most importantly, baseline symptom levels were the strongest predictors of subsequent mental health outcomes for both disorders, underscoring the critical importance of early identification.

From a developmental perspective, this pattern is consistent with models of perinatal psychological adjustment under sustained contextual stress, wherein elevated baseline symptoms reflect cumulative psychosocial burden rather than transient distress. Longitudinal studies conducted under conditions of heightened adversity suggest that chronic stress exposure constrains adaptive recovery across the perinatal transition, resulting in symptom persistence rather than spontaneous remission (La Rosa et al., 2024, 2025. In such contexts, psychosocial resources such as social support may function less as modifiers of risk onset and more as critical determinants of symptom trajectories over time. Our findings align with this framework, indicating that biological vulnerabilities unfold within broader stress-laden environments that shape perinatal mental health adjustment across pregnancy and the postpartum period.

Perceived social support was a robust independent protective factor and was statistically associated with the relationship between hemoglobin levels and depressive symptoms, accounting for ~41% of the total association in exploratory analyses. Collectively, these results underscore the critical importance of a biopsychosocial framework that acknowledges the interplay between biological deficiencies and psychosocial resources in shaping perinatal mental health (Borrell-Carrió et al., 2004).

Our findings contribute to a growing corpus of evidence from South Asia indicating that the burden of perinatal depression and anxiety in Pakistan is substantially higher than in high-income countries (Fisher et al., 2012). The prevalence we observed aligns with large meta-analyses reporting antenatal depression around 37% and postnatal depression around 30% in Pakistan (Atif et al., 2021; Waqas et al., 2023). Although less frequently studied, anxiety has been documented in up to 30% of perinatal women in similar cohorts (Bauer et al., 2024). Our study extends this literature by examining both conditions concurrently and tracking their trajectories, thereby demonstrating the dynamic and persistent nature of mental health morbidity across the perinatal period.

Beyond Pakistan, similar patterns have been documented across other low- and middle-income countries, where perinatal mental disorders are shaped by intersecting biological vulnerabilities and chronic psychosocial stressors (Bauer et al., 2024). Studies from South Asia, Sub-Saharan Africa and Latin America report comparably elevated rates of antenatal and postnatal depression and anxiety, often exceeding those observed in high-income settings, particularly where poverty, food insecurity, gendered stress and limited access to mental health services coexist (Fisher et al., 2012). Evidence from LMIC cohorts also suggests that nutritional deficiencies and social adversity frequently co-occur, reinforcing vulnerability to persistent or recurrent symptoms

across the perinatal period (Lassi et al., 2021; Roddy Mitchell et al., 2023). Situating our findings within this broader LMIC literature underscores that the observed symptom trajectories and context-specific biomarker associations likely reflect structural conditions common across resource-constrained settings rather than country-specific anomalies.

The biological findings highlight nutritional deficiencies as significant, yet often overlooked, contributors to perinatal psychopathology in LMICs. The protective effect of hemoglobin against both depression and anxiety is biologically plausible. Anemia can induce fatigue, cognitive impairment and diminished stress resilience, which may lower the threshold for developing mood and anxiety disorders (Wassef et al., 2019). While few studies in Pakistan have empirically tested this link, our findings indicate that hemoglobin levels are associated with cross-sectional symptomatology during pregnancy; however, these associations did not persist in adjusted longitudinal models once baseline symptom severity was accounted for. This may have clinical relevance, as hemoglobin testing could prompt heightened awareness of potential vulnerability when considered alongside psychosocial and clinical factors (Zeng et al., 2022).

Vitamin D presented a more complex, time-dependent relationship. Its antenatal deficiency was explicitly associated with anxiety, whereas postnatal deficiency emerged as a strong risk factor for depression. Furthermore, the trajectory analysis provided a crucial insight: women who were depressed during pregnancy but recovered postpartum (the "Remitted" group) were significantly more likely to be vitamin D sufficient compared to those with persistent or postnatal-onset depression. This temporal specificity is supported by mechanistic research indicating that vitamin D regulates neuro-inflammation and neuro-steroid synthesis processes that are particularly salient during the profound immune and hormonal shifts of the postpartum period (Accortt et al., 2022). Our results suggest a distinct role in anxiety, meriting further investigation. Given that vitamin D deficiency is endemic among Pakistani women of reproductive age (Etyemez et al., 2024), these findings highlight the potential relevance of micronutrient status for perinatal mental health and support further investigation into integrated screening approaches. Vitamin D deficiency in the depressed group and nondepressed group was not statistically significant ($p = 0.076$). This finding suggests that vitamin D deficiency, while endemic (Mahar et al., 2024b), is not a sufficient standalone cause of depression. It appears that the nondepressed women possessed compensatory protective factors. This pattern underscores a multidimensional model of resilience, where robust psychosocial and biological resources can buffer against the psychological impact of specific nutritional deficiencies, aligning with the biopsychosocial model (Borrell-Carrió et al., 2004).

The psychosocial dimensions of our results are equally critical. We confirmed that social support is a strong buffer against perinatal depression and anxiety, corroborating both international (Surkan et al., 2006; O'Hara and McCabe, 2013) and local evidence (Atif et al., 2021; Waqas et al., 2024). More importantly, we advanced the biopsychosocial model by demonstrating that perceived social support was statistically associated with the relationship between hemoglobin levels and depressive symptoms. This pattern suggests a potential psychosocial mechanism through which biological vulnerability and depressive symptoms may co-occur; anemia may impair a woman's capacity for social engagement, thereby eroding access to protective relationships and increasing depression risk (Grey et al., 2023). This insight has significant public health relevance in contexts like Pakistan, where postpartum social

isolation and familial conflict can exacerbate mental health vulnerabilities (Bedaso et al., 2021; Nazir et al., 2022b).

From a global mental health perspective, these findings support consideration of integrated interventions. While much international biomarker research has concentrated on neuroendocrine pathways (Glynn and Sandman, 2014), our results indicate that more common and modifiable nutritional factors warrant greater attention in LMICs. Similarly, despite strong evidence for their efficacy, psychosocial interventions remain under-resourced (Rahman et al., 2014). An integrated model combining nutritional support with culturally tailored psychosocial interventions (including social support), perhaps delivered by community health workers, could represent a promising area for future intervention research to alleviate the burden of perinatal mental illness in Pakistan and similar settings (Sikander et al., 2019).

## Limitations

This study has several limitations. Approximately 32% of participants were lost to follow-up, which reduced statistical power for longitudinal, trajectory and mediation analyses and may have limited the detection of small effects. Although formal attrition analyses indicated no statistically significant differences in baseline sociodemographic characteristics, biomarkers, perceived social support or mental health symptom severity between participants retained at follow-up and those lost to follow-up, the possibility of residual attrition bias cannot be fully excluded.

The sample was recruited from a single urban public-sector hospital, which may limit generalizability to rural settings or women receiving care in private facilities. In addition, while hemoglobin and vitamin D were included as biological markers, other potentially relevant biological pathways, such as inflammatory or neuroendocrine processes, were not assessed longitudinally. Finally, exploratory mediation analyses were based on baseline-only measurements, precluding temporal ordering and causal inference.

Notwithstanding these limitations, this study provides rare longitudinal evidence from Pakistan that jointly examines biological and psychosocial correlates of perinatal depression and anxiety. The findings demonstrate that hemoglobin and vitamin D show context- and stage-specific associations with symptom patterns, while perceived social support emerges as a consistently protective psychosocial factor. Together, these results strengthen understanding of how nutritional status and social context shape perinatal mental health in low-resource settings and underscore the importance of integrating biological and psychosocial perspectives in future research and maternal health planning.

## Implications for practice and policy

Our findings have important implications for clinical practice and maternal health policy in low- and middle-income countries. Given that hemoglobin and vitamin D levels are routinely assessed, abnormal values could serve as early indicators to prompt psychosocial evaluation, including brief screening or referral within antenatal care (Zhang et al., 2020). The strong association with perceived social support highlights the need for integrated care models, as biological and psychosocial interventions appear to operate synergistically. Nutritional supplementation alone is unlikely to be effective without addressing social isolation, and vice versa. Existing community-based interventions in Pakistan (Rahman et al., 2014; Sikander et al., 2019) could be strengthened by incorporating structured nutritional components. The high

attrition observed in the postpartum phase reflects systemic gaps in follow-up care, underscoring the need for routine postpartum visits that address both nutritional status and psychological well-being, potentially through task-sharing with trained lay health workers. Future Research DirectionsFuture research should explicitly differentiate between depression and anxiety, given their distinct biomarker associations observed in this study. Larger, multisite longitudinal studies are needed to validate these findings and develop integrated predictive models incorporating nutritional, endocrine, inflammatory, and psychosocial factors.

**Open peer review.** To view the open peer review materials for this article, please visit http://doi.org/10.1017/gmh.2026.10184.

**Data availability statement.** The dataset generated and analyzed during this study contains sensitive personal health information and cannot be made openly available. De-identified data may be shared upon reasonable request. Requests for access to the data should be directed to Dr. Abid Malik (Email: abid.malik@hsa.edu.pk).

**Acknowledgments.** The authors would like to thank the participating mothers for their time and trust in contributing to this research. We also extend our appreciation to the field research team and data collection staff for their dedicated efforts. We are grateful to the staff of the Federal General Hospital, Islamabad, for their cooperation throughout recruitment and data collection.

**Author contribution.** Dr. Abid Malik conceptualized the study, supervised the overall project and reviewed the initial manuscript draft. Rakhshanda Liaqat conducted the data analysis and wrote the manuscript. Nadia Suleman coordinated field activities and provided project management. Bilal Ahmed and Raima Asif contributed to manuscript proofreading. Shahzad Ali Khan, Pamela J. Surkan and Lauren M. Osborne critically reviewed the manuscript and provided intellectual input to strengthen the interpretation of findings. All authors reviewed and approved the final version of the manuscript before submission.

**Financial support.** This study was supported by the International Brain Research Organization (IBRO) through the Neuroscience Capacity Accelerator for Mental Health, awarded to Dr. Abid Malik (Application ID: IW-6827570252). No additional external funding was received for this research.

**Competing interests.** The authors declare no conflicts of interest.

**Ethical approval statement.** Ethical approval for this study was obtained from the National Bioethics Committee of Pakistan, Islamabad, and the Institutional Review Board (IRB) of the Health Services Academy, Islamabad. Written informed consent was obtained from all participants before data collection. For participants with low literacy, the consent form was read aloud in the presence of a literate witness, and verbal consent was recorded.

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
