## [Reviewer Report]

I appreciated the opportunity to review this manuscript. The study addresses a critical public health issue in a context where high-quality longitudinal data are scarce. The attempt to integrate nutritional biomarkers with psychosocial resources is both timely and conceptually sound. The manuscript is well-written and clearly structured. However, several points require careful revision to enhance methodological rigor, interpretive accuracy, and alignment between results and conclusions.

Major Revisions

1. Although the initial sample size calculation is clearly described, the substantial loss to follow-up (approximately 32%) raises important concerns that are not adequately addressed. The manuscript states that attrition was largely unrelated to symptom severity; however, no empirical comparison between retained and lost participants is reported. A formal attrition analysis comparing baseline demographic, clinical, biomarker, and mental health variables between completers and non-completers should be included. Without this analysis, the validity of the longitudinal GLMM findings remains uncertain. Additionally, the reduced sample size at follow-up substantially limits the power of trajectory and mediation analyses, and this limitation should be explicitly acknowledged.

2. The cross-sectional logistic regression models appear to be largely unadjusted, yet the results are sometimes interpreted as independent effects. The rationale for selecting covariates should be clarified, and key potential confounders, such as socioeconomic status, parity, education, BMI, and family structure, should be considered or explicitly justified if excluded. Similarly, the longitudinal models are only adjusted for baseline symptom severity, gestational age, and biomarkers. Clearer justifications are needed for excluding psychosocial variables, such as social support, from the primary GLMMs, given their central theoretical role in the study.

3. The mediation analysis that examines social support as a mediator of the hemoglobin–depression association is interesting from a conceptual standpoint, but it is methodologically fragile. Since all variables used in the mediation model were measured at baseline, it is not possible to establish temporal precedence among exposure, mediator, and outcome. Therefore, the use of causal mediation terminology should be substantially toned down throughout the manuscript. The authors should clearly frame this analysis as exploratory and associative and explicitly discuss the strong assumptions required for causal interpretation, including the absence of unmeasured confounding factors.

4. The trajectory groups, particularly those for anxiety, include very small cell sizes, which limits interpretability and statistical reliability. Although Fisher’s exact tests are appropriate, the manuscript should more clearly communicate the exploratory nature of these analyses and avoid drawing strong conclusions from subgroup comparisons with very small sample sizes. This is especially important when linking vitamin D status to remission or persistence patterns.

5. The role of hemoglobin and vitamin D is interpreted inconsistently across cross-sectional, longitudinal, and trajectory analyses. For instance, hemoglobin is considered a meaningful protective factor; however, it loses significance in adjusted longitudinal models. These findings should be more clearly reconciled in the discussion, emphasizing that baseline symptom severity overwhelmingly explains postnatal outcomes and that the effects of biomarkers appear context-specific and weaker once prior mental health is accounted for.

6. Although the policy and clinical implications are appealing, some of the recommendations currently exceed what the data can robustly support. Statements suggesting that biomarkers can be used for mental health risk stratification should emphasize that the findings are preliminary and hypothesis-generating, not practice-changing.

7. To strengthen the discussion on the interaction between psychosocial resources, stress, and perinatal mental health trajectories, it would be helpful to incorporate literature examining how contextual stressors influence perinatal psychological adjustment over time, especially during periods of increased stress. Studies on perinatal mental health under adverse contextual conditions could situate the present findings within a broader developmental and psychosocial framework (e.g., 10.3389/fpsyg.2025.1588433; 10.1007/s12144-024-06603-3.

Minor Revisions

1. Please ensure consistent use of terms such as perinatal, antenatal, and postnatal throughout the manuscript, particularly in the Results and Discussion sections.

2. It would be helpful to specify if biomarker assays were repeated for quality control purposes, as well as how missing data was handled beyond the mention of a complete-case approach.

3. Although several limitations are acknowledged, this section should explicitly include issues related to attrition bias, residual confounding, and constraints of baseline-only mediation analyses.

4. The overall language quality is good, but there are occasional instances of overly strong causal phrasing that should be softened. Further improving readability would require minor grammatical edits and sentence streamlining.

---

## [Reviewer Report]

Dear Author,

I found the study clear and engaging, and results are potentially valuable to the field. I have few suggestions to strengthen the manuscript.

Abstract: the abstract is clear however a brief clarification of the rationale for focusing on Hb and Vit D may help the readers who are not familiar with regional nutritional context.

Multiple approaches are mentioned, please add short indication of primary analytical emphasis (e.g longitudinal vs cross sectional) to improve clarity.

Introduction:

Minor streamlining of background content could improve flow and reduce repetition for readers familiar with this field.

Methodology:

To address the possibility of attrition-related bias, it would be beneficial to briefly mention whether there were differences in baseline characteristics between the retained (follow-up) and lost (follow-up) participants.

Results:

some numerical results are repeated across the text and tables, minor consolidation can improve readability of content.

Conclusion

Well supported by findings.

Good Luck...!!

Regards

Dr. Ayesha Sana

Sr. Lecturer

Pharmacy Dept

Iqra University, H-9 campus Islamabad

---

## [Reviewer Report]

Thank you for the opportunity to review this manuscript. It presents a longitudinal investigation of biopsychosocial stressors and their influence on perinatal mental health outcomes, an important contribution to the literature from a low‑ and middle‑income country perspective. The manuscript is generally well written and thoughtfully structured. However, some minor issues require attention to improve clarity and coherence. The following suggestions are offered to support further improvement

Introduction

• Page 5, line 25: The referent for “they” is unclear and should be specified.

• The rationale for studying is strong. The inclusion of biological markers and a longitudinal design appropriately strengthens the justification.

Methods

• Indicate the period during which data were collected, especially since the discussion references post COVID recovery and contextual stressors in the study setting.

Results

• Page 12, line 38: Capitalize the first letter of the sentence.

• Page 14, line 33: Capitalize the first letter of the sentence.

Discussion

• Expand the comparison of findings to include evidence from other LMICs beyond Pakistan.

Implications

• Page 18, line 51: Replace “lady” with “lay.”

• The statement “Future studies should distinguish between depression and anxiety” needs clearer elaboration.

• Given the mean gestational age of participants, the findings underscore the importance of early screening in pregnancy, as baseline symptoms strongly influence later outcomes.

General Comments

• The manuscript contains some typographical errors that require correction.

---

## [Editor Report]

Dear Authors,

I am happy to share with you that we have independent reviews for your manuscript, which are positive. However, there are some suggestions and observations as well, which I believe can enhance the scope of your study. Therefore, I request you address them and submit at the earliest possible.

With thanks and best wishes,

Thomas

---

## [Reviewer Report]

The manuscript has clearly benefited from the revision process. The structure is clearer, the results are well presented, and the discussion appropriately reflects the findings. The study is now suitable for publication. I recommend acceptance in its present form.

---

## [Editor Report]

Dear Authors,

I am happy to share with you that your manuscript has been accepted. I greatly appreciate you for revising the manuscript satisfactorily in time. I also thank you for considering the ‘Cambridge Prisms: Global Mental Health’ to present your work.

I look forward to your contributions to the journal in future.

Sincerely,

Thomas